# Genome-wide Identification of WRKY transcription factor family members in sorghum (*Sorghum bicolor* (L.) moench)

Elamin Hafiz Baillo[1,2,3,4]*, Muhammad Sajid Hanif[1,2], Yinghui Guo[1,2], Zhengbin Zhang[1,2,3]*, Ping Xu[1,2,3], Soad Ali Algam[5]

1 Center for Agricultural Resources Research, Key Laboratory of Agricultural Water Resources, Hebei Laboratory of Agricultural Water Saving, Institute of Genetics and Developmental Biology, University of Chinese Academy of Sciences, Shijiazhuang, Hebei, China, 2 University of Chinese Academy of Sciences, Beijing, China, 3 Innovation Academy for Seed Design, Chinese Academy of Sciences, Beijing, China, 4 Agricultural Research Corporation (ARC), Ministry of Agriculture, Wad Madani, Gezira, Sudan, 5 Faculty of Agriculture, University of Khartoum, Khartoum, Sudan

* aminomooon14@gmail.com (EHB); zzb@sjziam.ac.cn (ZZ)

**Data Availability Statement:** All relevant data are within the manuscript and its Supporting Information files.

## Abstract

WRKY transcription factors regulate diverse biological processes in plants, including abiotic and biotic stress responses, and constitute one of the largest transcription factor families in higher plants. Although the past decade has seen significant progress towards identifying and functionally characterizing *WRKY* genes in diverse species, little is known about the WRKY family in sorghum (*Sorghum bicolor* (L.) moench). Here we report the comprehensive identification of 94 putative WRKY transcription factors (*Sb*WRKYs). The *Sb*WRKYs were divided into three groups (I, II, and III), with those in group II further classified into five subgroups (IIa–IIe), based on their conserved domains and zinc finger motif types. WRKYs from the model plant Arabidopsis (*Arabidopsis thaliana*) were used for the phylogenetic analysis of all *SbWRKY* genes. Motif analysis showed that all *Sb*WRKYs contained either one or two WRKY domains and that *Sb*WRKYs within the same group had similar motif compositions. *SbWRKY* genes were located on all 10 sorghum chromosomes, and some gene clusters and two tandem duplications were detected. *SbWRKY* gene structure analysis showed that they contained 0–7 introns, with most *SbWRKY* genes consisting of two introns and three exons. Gene ontology (GO) annotation functionally categorized SbWRKYs under cellular components, molecular functions and biological processes. A *cis*-element analysis showed that all *SbWRKYs* contain at least one stress response-related *cis*-element. We exploited publicly available microarray datasets to analyze the expression profiles of 78 *SbWRKY* genes at different growth stages and in different tissues. The induction of *SbWRKYs* by different abiotic stresses hinted at their potential involvement in stress responses. qRT-PCR analysis revealed different expression patterns for *SbWRKYs* during drought stress. Functionally characterized *WRKY* genes in *Arabidopsis* and other species will provide clues for the functional characterization of putative orthologs in sorghum. Thus, the present study delivers a solid foundation for future functional studies of *SbWRKY* genes and their roles in the response to critical stresses such as drought.

**Funding:** This project was supported by The National Key Research and Development Program of China (2016YFD0100605), the Innovation Academy for Seed Design, Chinese Academy of Sciences; special exchange programme of the Chinese academy of sciences (category A).The authors sincerely thank the Center for Agricultural Resources Research, Institute of Genetics and Developmental Biology (CAS) for providing the facilities'.

**Competing interests:** The authors have declared that no competing interests exist.

## Introduction

WRKY transcription factors (TFs), one of the largest TF families in plants, regulate various biological processes, including stress responses. WRKY proteins contain a conserved WRKYGQK motif at their N-terminus, along with a 60-amino-acid-long zinc finger motif at their C-terminus [1]. These two motifs are essential for the binding of WRKY TFs to the W-box *cis*-element [(T)TGAC(C/T)] located within the promoters of their target genes. WRKY proteins can be classified into three groups (I, II, and III) according to the number of WRKY domains and the type of zinc finger motif, i.e., $C_2H_2$ or $C_2HC$ [2]. Group I members have two WRKY domains and the $C_2H_2$-type zinc finger. Group II members have only one WRKY domain and a $C_2H_2$ zinc finger motif and can be further classified into five subgroups (IIa, IIb, IIc, IId, and IIe) based on the sequence of the DNA-binding domain. Finally, group III members have one WRKY domain and a $C_2HC$-type zinc finger [3, 4].

Several studies have demonstrated that WRKY TFs regulate various biological processes and control gene expression via a combination of positive or negative regulation [5, 6]. WRKY TFs have been reported to be involved in responses to biotic stresses [7], developmental processes such as senescence, embryogenesis, and seed development, as well as abiotic stresses [8, 9]. For example, the wheat (*Triticum aestivum*) *TaWRKY10* gene is considered to be a key regulator in salt and drought responses by regulating stress-responsive genes [10]. Heterologous expression of *TaWRKY1* and *TaWRKY33* enhanced drought and heat tolerance in *Arabidopsis* plants [11]. Also, *Arabidopsis* plants heterologously expressing the maize (*Zea mays*) *ZmWRKY40* gene exhibited improved tolerance to drought [12]. Similarly, heterologous expression of *TaWRKY13* in *Arabidopsis* increased root length and proline content, and reduced malondialdehyde content, thus improving salt stress tolerance [13]. Overexpression of *WRKY13* in Arabidopsis enhanced cadmium tolerance of transgenic plants by inducing the expression of *PLEIOTROPIC DRUG RESISTANCE 8* (*PDR8*), encoding an ATP-binding cassette transporter [14].

*WRKY* can however also act as a negative regulator of gene expression. Heterologous expression of the cotton (*Gossypium hirsutum*) *GhWRKY33* gene reduced drought tolerance of transgenic Arabidopsis plants [15]. Likewise, heterologous expression of *ZmWRKY17* impaired salt stress tolerance in transgenic Arabidopsis and lowered the abscisic acid (ABA) content by repressing ABA-dependent and stress-responsive genes [16].

Beyond the functional characterization of *WRKY* genes in Arabidopsis, the functions of many *WRKY* genes remain to be validated in non-model species. Indeed, few studies have tested the contribution of *WKRY* genes in their species of origin. Overexpression of *TaWRKY2* in wheat enhanced tolerance to drought stress and increased yield [17]. Overexpression of *OsWRKY11* enhanced tolerance to drought and heat in transgenic rice [18]. The grapevine (*Vitis amurensis*) *VaWRKY12* gene enhanced cold tolerance in transgenic grapevine calli [19]. In wild sorghum (*Sorghum propinquum*), *SpWRKY* controls seed shattering but is unrelated to seed shattering genes selected during domestication, as it likely arose recently [20]. These studies confirm that WRKY TFs play important roles and suggest their potential use for crop improvement in terms of stress tolerance.

Since the identification of the first *WRKY* gene from sweet potato (*Ipomoea batatas*) [21], genome-wide analyses in different species have identified many *WRKY* genes, including 171 *WRKY* genes in wheat [22], 119 in maize [23], 103 in rice (*Oryza sativa*) [24], 71 in sesame (*Sesamum indicum*) [25], 70 in chickpea (*Cicer arietinum*) [26], and 79 in potato (*Lycopersi-cum tuberosum*) [27]. Sorghum is the fifth most important cereal crop in terms of production and dedicated arable land, and displays unique adaptations that allow it to withstand harsh conditions at different growth stages. Sorghum is also an excellent model for TF studies [28–30]. The availability of a complete genome assembly for sorghum now provides an opportunity

for the genome-wide identification of *SbWRKY* genes. To gain insight into the roles of SbWRKYs in plant responses to stresses such as drought, we used a variety of approaches to identify and functionally characterize 94 putative members of the WRKY family in sorghum.

## Material and methods

### Identification of WRKY family genes in sorghum

We collected data from the following databases to identify putative *SbWRKY* genes. The Plant Transcription Factor Database version 4 (http://planttfdb.cbi.pku.edu.cn/) was used to download the amino acid sequences of sorghum WRKY proteins, and "WRKY" was used as a query to search against the Grassius Transcription Factor Database (https://grassius.org/grasstfdb. php). We used the WRKY domain ID (PF03106) to identify putative WRKY proteins encoded by the *S. bicolor* genome (v3.1) through the Joint Genome Institute (JGI) (https://phytozome. jgi.doe.gov/pz/portal.html#). We also used the keyword "WRKY" as a search query in the MOROCOSHI Sorghum Transcription Factor Database (http://sorghum.riken.jp/morokoshi/ Home.html).

We employed CD-HIT suite (http://weizhong-lab.ucsd.edu/cdhit_suite/cgi-bin/index.cgi? cmd=cd-hit) to remove redundant and incomplete sequences, and Simple Modular Architecture Research Tool (SMART) (http://smart.embl-heidelberg.de/#) [31] to confirm that the sequences contained WRKY domain(s).

For all identified *Sb*WRKY proteins, we obtained their predicted isoelectric point (pI) and molecular weight (MW) from the ExPASy proteomic server (http://web.expasy.org/ protparam).

### Chromosome mapping and tandem duplications of *SbWRKY* genes

Information about the chromosomal locations of all identified *SbWRKY* genes was obtained using the Phytozome BioMart tool (https://phytozome.jgi.doe.gov/biomart/martview/), and the genes were mapped onto sorghum chromosomes by MapChart (v.2.32). Tandem duplications of *SbWRKYs* were based on the following criteria: genes within a 100-kbp region on an individual chromosome with a sequence similarity $\geq$ 70% [32]. We calculated sequence similarities using EMBOSS Water, which uses the Smith-Waterman local pairwise alignment algorithm (https://www.ebi.ac.uk/Tools/psa/emboss_water/).

### Classification and phylogenetic analysis of *SbWRKY* genes

*Arabidopsis* WRKY amino acid sequences obtained from the *Arabidopsis* Information Resource (TAIR) (https://www.arabidopsis.org/)) together with our *Sb*WRKY sequences were used to construct a phylogenetic tree and classify *SbWRKY* genes. We performed multiple sequence alignments with ClustalW for *At*WRKY and *Sb*WRKY protein sequences and constructed a phylogenetic tree using MEGA v7.0 (https://www.megasoftware.net/) and the p-distance model. Pairwise deletion and 1,000 bootstrap replicates were used for the neighbour-joining method [33]. Based on the phylogenetic tree of *At*WRKY and *Sb*WRKY sequences, *SbWRKY* genes were classified into different groups and subgroups. The identification of putative sorghum WRKY orthologs in *Arabidopsis* was based on sequence alignment data and the phylogenetic tree [27].

### Gene structure analysis of *SbWRKY* genes

The exon-intron structure of *SbWRKYs* was analyzed using the Gene Structure Display Server (GSDS v. 2.0) (http://gsds.cbi.pku.edu.cn/) from the Center for Bioinformatics at Peking

University [34]. The genomic sequence and coding sequence (CDS) of each *SbWRKY* gene were used to predict the exon-intron pattern.

## Conserved motif distribution analysis of *SbWRKY* genes

Conserved motifs in the *Sb*WRKY proteins were identified using Multiple Em for Motif Elicitation (MEME) (v. 4.9.0; http://meme.nbcr.net/meme/)) with the following parameters: maximum motif number: 20; site distributions: any number of repetitions; minimum and maximum width: 10 and 50, respectively [35].

## Gene ontology annotation and analysis of *cis*-acting elements

The gene ontology (GO) annotation analysis for *Sb*WRKY proteins was performed using the Blast2GO tool with default parameters [36]. We screened protein sequences using the Basic Local Alignment Tool for Proteins (BlastP), followed by functional analysis, including mapping and annotation. Moreover, we collected information related to the biological process, molecular function, and cellular component associated with each *Sb*WRKY. We also analyzed the *cis*-acting elements of *SbWRKY* genes by extracting 1,500 bp of upstream region for all *SbWRKY* genes and running the sequences through the online website PlantCARE (http://bioinformatics.psb.ugent.be/webtools/plantcare/html/).

## Digital expression pattern analysis of *SbWRKY* genes

To survey *SbWRKY* expression profiles, Affymetrix transcriptomic array data for sorghum were obtained from the Sorghum Functional Genomics Database (http://structuralbiology.cau.edu.cn/sorghum/pattern.php). The Genevestigator platform was used to analyze the expression profiles of *SbWRKYs* under different environmental stresses (drought, salt, ABA, and heat) with different samples stored in the Genevestigator platform [37]. The *SbWRKY* expression profiles heatmaps were generated using a hierarchical clustering analysis tool on the Genevestigator platform.

## Drought stress treatment and samples collection

Seeds of *Sorghum bicolor* genotype (SX44B) used in this study were provided by Professor Zhang Fuyao (Sorghum Institute, Shanxi Academy of Agricultural Sciences, Shanxi, China). The seeds were surface-sterilized with 75% (v/v) ethanol and 5% (v/v) sodium hypochlorite for 1–2 min, and then rinsed them three times with distilled water. Sterilized seeds were allowed to germinate on two layers of water-soaked paper and incubated at 25°C in darkness for 3 d. Seedlings were transferred to pots containing a soil mixture of vermiculite and peat moss in a 1:1 ratio. Seedlings were kept in a growth chamber with 60–70% relative humidity, 28°C/23°C day/night temperature cycles, and a 16-h light/8-h dark photoperiod. Seedlings were maintained under normal growth conditions for two weeks before exposure to drought treatment, seedlings were treated with PEG8000 20% [38]. The seedlings shoot samples were collected at 0, 3, 6, 12, and 24 h, all samples were frozen in liquid nitrogen, and then stored at –80°C until used for RNA extraction.

## Gene expression analysis by quantitative Real-Time PCR (qRT-PCR)

RNAs were extracted from the samples using (Promega, China) according to the manufacturer's specifications. The first-strand cDNAs were synthesized by reverse transcription of 100 μg total RNA which was generated using the Easy Script First-strand cDNA Synthesis SuperMix Kit (TransGen Biotech, China). Synthesized cDNA was diluted 1:10 with nuclease-free water for use in qRT-PCR. The expression levels of the genes were normalized to the sorghum

housekeeping gene *GLYCERALDEHYDE 3-PHOSPHATE DEHYDROGENASE* (*GAPDH*) gene as an internal control. Gene-specific primers were designed using the National Center for Biotechnology Information (NCBI) Primer-BLAST tool and were synthesized by Sangon (Beijing, China). Primers used in this study are listed in Supplementary S7 File. qRT-PCR was carried out using *TransStart* Green qPCR SuperMix UDG (TransGen Biotech, China) following the manufacturer's instructions, on a Bio-Rad CFX96TM real-time PCR detection system (Bio-Rad, USA). Reaction parameters for thermal cycling were as follows: 94°C for 10 min, 40 cycles of 94°C for 5 sec and 55°C for 15 sec, 72°C for 10 sec. We performed RT-qPCR on three biological replicates and used the $2^{-\Delta\Delta Ct}$ method for quantification [39].

## Results

### Identification of WRKY family members in sorghum

Taking advantage of the availability of a complete genome assembly for sorghum, we identified *Sb*WRKY family members using the keyword "WRKY" and the WRKY domain consensus sequence (PF03106) as queries in different databases. In this study, the presence of the WRKYGQK or WRKYGQK-like conserved domain was the basic criterion for the inclusion of genes in the *Sb*WRKY family. We initially identified 134, 94, 99, and 97 transcripts from the TFDB, Grassius, MOROCOSHI, and JGI databases, respectively. We then used CD-HIT and multiple sequence alignments to remove redundant *Sb*WRKY protein sequences and confirmed the presence of the WRKY domain in the remaining sequences by running all proteins through the SMART database. We thus removed all redundant sequences and those with an incomplete WRKY domain. A total of 94 non-redundant *Sb*WRKY sequences were identified, and the following gene and protein data were summarized in Table 1. We named *SbWRKYs* according to their physical positions along sorghum chromosomes, starting with the upper arm of chromosome 1 and moving down to the lower arm (*SbWRKY1* to *SbWRKY94*), as described previously [1] (Table 1). We used this set of *Sb*WRKY protein sequences for further characterization. *Sb*WRKY proteins ranged from 110 to 1,584 amino acids, with an average length of 390 amino acids. Their predicted MW and pI values ranged from 12.23 to 78.24 and from 4.75 to 10.06, respectively.

### Chromosome mapping of the *SbWRKY* genes and tandem duplication analysis

MapChart was used to determine the locations of the 94 *SbWRKY* genes, and they were distributed on all 10 sorghum chromosomes (Chr) (Fig 1). Chr 3 counted the highest number of *SbWRKYs* with 23 genes, corresponding to 24.5% of the entire gene family, followed by 14 genes on Chr 9, 11 genes on Chr 1, and 10 genes on Chr2. Chromosomes 6, 7, and 10 had the fewest number of genes, with only five *SbWRKYs* each. The remaining *SbWRKY* genes were located on Chr4 (six genes), Chr5 (seven genes), and Chr8 (eight genes).

Tandem repeats were identified based on previously reported criteria [40]; namely, two or more genes should be located within a 100-kbp window and display a sequence similarity of at least 70%. Gene cluster events were observed on six chromosomes. Specifically, there were five clusters on Chr3; two each on Chr2 and Chr9; and one cluster each on Chr1, Chr5, and Chr8. No clusters were found on Chr4, Chr6, Chr7, and Chr10. High-density clusters were detected on Chr5 and Chr3, and identified tandem repeats on two chromosomes: two group III genes (*SbWRKY51* and *SbWRKY52*) on Chr5 and two group-IIc genes (*SbWRKY87* and *SbWRKY88*) on Chr9 (Fig 1). These tandem-duplicated genes clustered together in the phylogenetic tree within their respective clades, and the sequence similarity metrics for the gene pairs are provided in S1 File.

**Table 1. Characteristics of the identified *SbWRKY* genes.**

| Gene Name | Gene Locus ID | Chromosome Location | Gene Start | Gene End | pI | MW | Conserved Heptapeptide | Zinc Finger Type | Domain Number | Group | Protein Length (aa) |
|---|---|---|---|---|---|---|---|---|---|---|---|
| *SbWRKY1* | Sobic.001G006600.1 | Chr01 | 661532 | 669834 | 6.10 | 42.50 | WRKYGQK | C2H2 | 1 | lle | 392 |
| *SbWRKY2* | Sobic.001G055400.1 | Chr01 | 4145000 | 4147281 | 9.42 | 46.08 | WRKYGQK | C2H2 | 1 | lle | 434 |
| *SbWRKY3* | Sobic.001G083000.1 | Chr01 | 6408709 | 6412370 | 6.76 | 45.19 | WRKYGQK | C2H2 | 2 | l | 424 |
| *SbWRKY4* | Sobic.001G084000.1 | Chr01 | 6506824 | 6511079 | 6.2 | 34.37 | WRKYGQK | C2H2 | 1 | llc | 331 |
| *SbWRKY5* | Sobic.001G095500.1 | Chr01 | 7340447 | 7343822 | 10.05 | 43.61 | WRKYGQK | C2H2 | 1 | lld | 406 |
| *SbWRKY6* | Sobic.001G148000.1 | Chr01 | 11929277 | 11931246 | 8.89 | 15.12 | WRKYGQK | C2H2 | 1 | NG | 141 |
| *SbWRKY7* | Sobic.001G162100.1 | Chr01 | 13349344 | 13352564 | 9.78 | 37.88 | WRKYGQK | C2H2 | 1 | lld | 352 |
| *SbWRKY8* | Sobic.001G282400.1 | Chr01 | 55413200 | 55415329 | 5.97 | 46.31 | WRKYGQK | C2H2 | 1 | lle | 427 |
| *SbWRKY9* | Sobic.001G332500.1 | Chr01 | 62096233 | 62103315 | 7.34 | 52.62 | WRKYGQK | C2HC | 2 | l | 498 |
| *SbWRKY10* | Sobic.001G381300.1 | Chr01 | 66930715 | 66931988 | 5.60 | 33.95 | WRKYGQK | C2HC | 1 | lll | 328 |
| *SbWRKY11* | Sobic.001G389000.1 | Chr01 | 67559616 | 67561361 | 8.71 | 24.20 | **WRKYGEK** | C2H2 | 1 | lll | 225 |
| *SbWRKY12* | Sobic.002G008600.2 | Chr02 | 791610 | 795938 | 9.71 | 29.42 | WRKYGQK | C2H2 | 1 | llc | 272 |
| *SbWRKY13* | Sobic.002G128400.2 | Chr02 | 17755739 | 17757847 | 5.75 | 29.09 | WRKYGQK | C2H2 | 1 | lll | 275 |
| *SbWRKY14* | Sobic.002G168300.3 | Chr02 | 52688090 | 52696535 | 6.92 | 17.74 | **WRKYGSK** | C2HC | 2 | lll | 1584 |
| *SbWRKY15* | Sobic.002G174200.1 | Chr02 | 55134836 | 55136822 | 5.67 | 32.89 | WRKYGQK | C2HC | 1 | lll | 310 |
| *SbWRKY16* | Sobic.002G174300.1 | Chr02 | 55188598 | 55190855 | 5.99 | 34.60 | WRKYGQK | C2HC | 1 | lll | 328 |
| *SbWRKY17* | Sobic.002G202700.1 | Chr02 | 59258884 | 59260622 | 6.46 | 34.72 | WRKYGQK | C2H2 | 1 | lla | 327 |
| *SbWRKY18* | Sobic.002G202800.1 | Chr02 | 59275303 | 59277597 | 6.72 | 31.77 | WRKYGQK | C2H2 | 1 | lla | 295 |
| *SbWRKY19* | Sobic.002G242500.1 | Chr02 | 63158788 | 63162729 | 6.94 | 64.54 | WRKYGQK | C2H2 | 1 | l | 602 |
| *SbWRKY20* | Sobic.002G355000.1 | Chr02 | 71806701 | 71813179 | 6.24 | 65.56 | WRKYGQK | C2H2 | 2 | l | 611 |
| *SbWRKY21* | Sobic.002G418500.1 | Chr02 | 76629585 | 76632298 | 9.17 | 37.05 | WRKYGQK | C2HC | 1 | lll | 354 |
| *SbWRKY22* | Sobic.003G000600.1 | Chr03 | 65678 | 68474 | 8.62 | 59.48 | WRKYGQK | C2H2 | 1 | llb | 570 |
| *SbWRKY23* | Sobic.003G037400.1 | Chr03 | 3517021 | 3519078 | 6.07 | 26.65 | **WRKYGKK** | C2H2 | 1 | llc | 260 |
| *SbWRKY24* | Sobic.003G037500.1 | Chr03 | 3520321 | 3522720 | 7.27 | 56.13 | WRKYGQK | C2H2 | 1 | llb | 548 |
| *SbWRKY25* | Sobic.003G040800.1 | Chr03 | 3787973 | 3794918 | 9.75 | 30.73 | WRKYGQK | C2H2 | 1 | llc | 295 |
| *SbWRKY26* | Sobic.003G138400.1 | Chr03 | 13400561 | 13405829 | 6.77 | 60.58 | WRKYGQK | C2H2 | 1 | llb | 582 |
| *SbWRKY27* | Sobic.003G199400.1 | Chr03 | 52648841 | 52650568 | 5.57 | 23.75 | **WRKYGKK** | C2H2 | 1 | llc | 216 |
| *SbWRKY28* | Sobic.003G200700.1 | Chr03 | 52845063 | 52846563 | 9.17 | 33.94 | WRKYGQK | C2H2 | 1 | llb | 332 |
| *SbWRKY29* | Sobic.003G226600.1 | Chr03 | 56293002 | 56295461 | 8.7 | 44.22 | WRKYGQK | C2H2 | 1 | lle | 413 |
| *SbWRKY30* | Sobic.003G227300.1 | Chr03 | 56472263 | 56479136 | 6.62 | 38.87 | WRKYGQK | C2H2 | 1 | llc | 361 |
| *SbWRKY31* | Sobic.003G242800.2 | Chr03 | 58197072 | 58201064 | 6.45 | 35.97 | WRKYGQK | C2HC | 1 | lll | 346 |
| *SbWRKY32* | Sobic.003G248400.1 | Chr03 | 58718991 | 58722963 | 7.32 | 42.36 | WRKYGQK | C2H2 | 1 | llc | 410 |
| *SbWRKY33* | Sobic.003G276000.1 | Chr03 | 61248940 | 61250384 | 8.64 | 23.95 | **WRKYGKK** | C2H2 | 1 | llc | 225 |
| *SbWRKY34* | Sobic.003G285500.1 | Chr03 | 61880213 | 61882652 | 5.27 | 32.23 | WRKYGQK | C2H2 | 1 | lle | 319 |
| *SbWRKY35* | Sobic.003G287200.1 | Chr03 | 62028638 | 62032127 | 7.08 | 26.67 | WRKYGQK | C2H2 | 1 | llc | 246 |
| *SbWRKY36* | Sobic.003G296300.1 | Chr03 | 62835017 | 62837081 | 4.75 | 33.69 | WRKYGQK | C2H2 | 1 | lle | 310 |
| *SbWRKY37* | Sobic.003G337500.1 | Chr03 | 66057329 | 66064893 | 5.85 | 26.64 | WRKYGQK | C2HC | 1 | lll | 236 |
| *SbWRKY38* | Sobic.003G337600.1 | Chr03 | 66070688 | 66076206 | 5.98 | 29.56 | WRKYGQK | C2HC | 1 | lll | 264 |
| *SbWRKY39* | Sobic.003G337700.1 | Chr03 | 66081053 | 66083714 | 5.87 | 40.92 | WRKYGQK | C2HC | 1 | lll | 377 |
| *SbWRKY40* | Sobic.003G337800.1 | Chr03 | 66089525 | 66092348 | 5.17 | 36.43 | WRKYGQK | C2HC | 1 | lll | 333 |
| *SbWRKY41* | Sobic.003G337900.1 | Chr03 | 66104667 | 66107290 | 5.91 | 29.63 | WRKYGQK | C2HC | 1 | lll | 277 |
| *SbWRKY42* | Sobic.003G341100.1 | Chr03 | 66392943 | 66395715 | 6.32 | 59.35 | WRKYGQK | C2H2 | 2 | l | 556 |
| *SbWRKY43* | Sobic.003G353000.1 | Chr03 | 67192266 | 67198503 | 6.13 | 39.68 | WRKYGQK | C2H2 | 1 | NG | 377 |
| *SbWRKY44* | Sobic.003G444000.1 | Chr03 | 74207255 | 74210127 | 4.76 | 38.85 | WRKYGQK | C2H2 | 1 | llc | 354 |
| *SbWRKY45* | Sobic.004G065900.1 | Chr04 | 5352084 | 5353418 | 8.45 | 38.95 | WRKYGQK | C2H2 | 1 | lla | 364 |
| *SbWRKY46* | Sobic.004G117600.1 | Chr04 | 12353902 | 12355528 | 5.81 | 39.57 | WRKYGQK | C2H2 | 1 | lle | 375 |

*(Continued)*

**Table 1.** (Continued)

| Gene Name | Gene Locus ID | Chromosome Location | Gene Start | Gene End | pI | MW | Conserved Heptapeptide | Zinc Finger Type | Domain Number | Group | Protein Length (aa) |
|---|---|---|---|---|---|---|---|---|---|---|---|
| *SbWRKY47* | Sobic.004G138400.2 | Chr04 | 38911863 | 38913035 | 9.78 | 32.03 | WRKYGQK | C2H2 | 1 | lld | 299 |
| *SbWRKY48* | Sobic.004G271800.1 | Chr04 | 61583724 | 61588753 | 5.67 | 51.61 | WRKYGQK | C2H2 | 1 | lle | 497 |
| *SbWRKY49* | Sobic.004G298400.1 | Chr04 | 63778460 | 63782424 | 8.43 | 26.10 | WRKYGQK | C2H2 | 1 | llc | 238 |
| *SbWRKY50* | Sobic.004G312200.1 | Chr04 | 64900147 | 64902898 | 5.09 | 61.31 | WRKYGQK | C2H2 | 1 | llb | 578 |
| *SbWRKY51* | Sobic.005G013400.1 | Chr05 | 1211003 | 1212487 | 5.69 | 36.49 | **WRKYGEK** | C2HC | 1 | lll | 334 |
| *SbWRKY52* | Sobic.005G013500.1 | Chr05 | 1224392 | 1225924 | 6.31 | 38.95 | **WRKYGEK** | C2HC | 1 | lll | 361 |
| *SbWRKY53* | Sobic.005G013600.2 | Chr05 | 1235265 | 1241922 | 8.85 | 25.24 | **WRKYGEK** | C2HC | 1 | lll | 227 |
| *SbWRKY54* | Sobic.005G013800.1 | Chr05 | 1245144 | 1246547 | 5.88 | 30.67 | WRKYGQK | C2HC | 1 | lll | 271 |
| *SbWRKY55* | Sobic.005G014000.1 | Chr05 | 1261347 | 1263314 | 6.27 | 33.04 | WRKYGQK | C2HC | 1 | lll | 289 |
| *SbWRKY56* | Sobic.005G014200.1 | Chr05 | 1302596 | 1304021 | 6 | 30.80 | WRKYGQK | C2HC | 1 | lll | 272 |
| *SbWRKY57* | Sobic.005G117400.2 | Chr05 | 51657060 | 51660442 | 9.18 | 24.24 | WRKYGQK | C2H2 | 1 | llc | 225 |
| *SbWRKY58* | Sobic.006G051700.1 | Chr06 | 38157163 | 38162677 | 5.97 | 12.23 | WRKYGQK | C2HC | 1 | lll | 110 |
| *SbWRKY59* | Sobic.006G115700.1 | Chr06 | 48356715 | 48362695 | 6.77 | 78.24 | WRKYGQK | C2H2 | 2 | l | 740 |
| *SbWRKY60* | Sobic.006G166300.1 | Chr06 | 52377828 | 52383758 | 8.66 | 29.10 | WRKYGQK | C2H2 | 1 | llc | 269 |
| *SbWRKY61* | Sobic.006G201000.1 | Chr06 | 55241047 | 55245365 | 7.01 | 55.79 | WRKYGQK | C2H2 | 1 | lle | 532 |
| *SbWRKY62* | Sobic.006G206000.1 | Chr06 | 55572908 | 55574447 | 10.05 | 33.13 | WRKYGQK | C2H2 | 1 | lld | 315 |
| *SbWRKY63* | Sobic.007G077466.1 | Chr07 | 8936871 | 8940145 | 7.7 | 49.41 | WRKYGQK | C2H2 | 1 | llc | 455 |
| *SbWRKY64* | Sobic.007G085300.1 | Chr07 | 10754655 | 10756113 | 9.87 | 32.97 | WRKYGQK | C2H2 | 1 | lld | 318 |
| *SbWRKY65* | Sobic.007G111600.1 | Chr07 | 41587770 | 41594141 | 6.05 | 61.78 | WRKYGQK | C2H2 | 2 | l | 569 |
| *SbWRKY66* | Sobic.007G118301.1 | Chr07 | 51192936 | 51194468 | 6.48 | 35.62 | WRKYGQK | C2HC | 1 | lll | 340 |
| *SbWRKY67* | Sobic.007G217700.3 | Chr07 | 64590499 | 64600679 | 6.89 | 74.52 | WRKYGQK | C2H2 | 2 | l | 685 |
| *SbWRKY68* | Sobic.008G028600.1 | Chr08 | 2541872 | 2544263 | 5.35 | 36.20 | WRKYGQK | C2HC | 1 | lll | 341 |
| *SbWRKY69* | Sobic.008G029000.2 | Chr08 | 2604929 | 2606770 | 6.72 | 33.68 | WRKYGQK | C2HC | 1 | lll | 294 |
| *SbWRKY70* | Sobic.008G029200.1 | Chr08 | 2612734 | 2614762 | 5.99 | 30.33 | WRKYGQK | C2HC | 1 | lll | 267 |
| *SbWRKY71* | Sobic.008G029400.1 | Chr08 | 2625199 | 2632242 | 8.94 | 25.19 | **WRKYGEK** | C2HC | 1 | lll | 225 |
| *SbWRKY72* | Sobic.008G060300.1 | Chr08 | 6462595 | 6464519 | 5.15 | 35.55 | WRKYGQK | C2HC | 1 | lll | 334 |
| *SbWRKY73* | Sobic.008G107500.1 | Chr08 | 50603514 | 50609969 | 7.71 | 52.42 | WRKYGQK | C2H2 | 2 | l | 496 |
| *SbWRKY74* | Sobic.008G153600.1 | Chr08 | 58584243 | 58587317 | 10.06 | 39.29 | WRKYGQK | C2H2 | 1 | lld | 371 |
| *SbWRKY75* | Sobic.008G174100.1 | Chr08 | 60851165 | 60856614 | 8.89 | 13.13 | **WRKSGQR** | C2HC | 1 | lll | 1163 |
| *SbWRKY76* | Sobic.009G034800.1 | Chr09 | 3182138 | 3188861 | 5.79 | 58.46 | WRKYGQK | C2H2 | 1 | llb | 567 |
| *SbWRKY77* | Sobic.009G068900.1 | Chr09 | 7567284 | 7568764 | 6.96 | 22.09 | **WRKYGKK** | C2H2 | 1 | llc | 206 |
| *SbWRKY78* | Sobic.009G092100.1 | Chr09 | 20912936 | 20928927 | 6.39 | 40.83 | WRKYGQK | C2HC | 1 | III | 378 |
| *SbWRKY79* | Sobic.009G100500.1 | Chr09 | 39826393 | 39831148 | 7.69 | 54.06 | WRKYGQK | C2H2 | 2 | l | 517 |
| *SbWRKY80* | Sobic.009G171600.1 | Chr09 | 52696414 | 52699630 | 8.28 | 68.54 | WRKYGQK | C2H2 | 2 | l | 649 |
| *SbWRKY81* | Sobic.009G174300.1 | Chr09 | 52963804 | 52966801 | 6.25 | 26.62 | **WRKYGEK** | C2HC | 1 | lll | 235 |
| *SbWRKY82* | Sobic.009G206800.1 | Chr09 | 55463596 | 55464452 | 9.2 | 25.32 | WRKYGQK | C2H2 | 1 | llc | 241 |
| *SbWRKY83* | Sobic.009G212800.1 | Chr09 | 55836119 | 55837874 | 6.08 | 22.71 | **WRKYGKK** | C2H2 | 1 | llc | 219 |
| *SbWRKY84* | Sobic.009G234100.1 | Chr09 | 57342319 | 57345920 | 6.38 | 44.44 | WRKYGQK | C2H2 | 1 | llc | 424 |
| *SbWRKY85* | Sobic.009G234900.1 | Chr09 | 57418453 | 57422639 | 6.52 | 65.84 | WRKYGQK | C2H2 | 1 | llb | 631 |
| *SbWRKY86* | Sobic.009G238200.1 | Chr09 | 57628850 | 57630763 | 5.81 | 29.08 | WRKYGQK | C2HC | 1 | lll | 272 |
| *SbWRKY87* | Sobic.009G247300.2 | Chr09 | 58261832 | 58265060 | 8.22 | 28.36 | WRKYGQK | C2H2 | 1 | llc | 262 |
| *SbWRKY88* | Sb09g029810 | Chr09 | 58261882 | 58265547 | 6.07 | 36.12 | WRKYGQK | C2H2 | 1 | llc | 343 |
| *SbWRKY89* | Sobic.009G247700.1 | Chr09 | 58309106 | 58310986 | 6.05 | 38.89 | WRKYGQK | C2H2 | 1 | lle | 364 |
| *SbWRKY90* | Sobic.010G035300.2 | Chr10 | 2858141 | 2865169 | 6.79 | 65.39 | WRKYGQK | C2H2 | 1 | llb | 625 |
| *SbWRKY91* | Sobic.010G045700.1 | Chr10 | 3566332 | 3570889 | 6.11 | 40.52 | WRKYGQK | C2HC | 1 | lll | 378 |
| *SbWRKY92* | Sobic.010G148600.2 | Chr10 | 42567454 | 42569300 | 6.01 | 41.69 | WRKYGQK | C2H2 | 1 | lle | 385 |

*(Continued)*

**Table 1.** (Continued)

| Gene Name | Gene Locus ID | Chromosome Location | Gene Start | Gene End | pI | MW | Conserved Heptapeptide | Zinc Finger Type | Domain Number | Group | Protein Length (aa) |
|---|---|---|---|---|---|---|---|---|---|---|---|
| *SbWRKY93* | Sobic.010G148800.1 | Chr10 | 42829856 | 42831874 | 7.58 | 37.43 | WRKYGQK | C2H2 | 1 | IIe | 350 |
| *SbWRKY94* | Sobic.010G209100.1 | Chr10 | 55261687 | 55263735 | 9.64 | 37.82 | WRKYGQK | C2H2 | 1 | IIa | 348 |

pI, isoelectric point; MW, molecular weight; aa, amino acid.

## Classification and phylogenetic analysis of *SbWRKYs*

To investigate the evolution of *Sb*WRKY family members, an unrooted phylogenetic tree was constructed based on multiple sequence alignment between full-length protein sequences of 65 *At*WRKYs and 94 *Sb*WRKYs, using the neighbour-joining method in MEGA7.0 (Fig 2). The constructed phylogenetic tree was used to classify the *Sb*WRKYs into three major groups (I, II, and III), according to the classification in *Arabidopsis* [1] (Fig 2). Of the 11 *Sb*WRKYs in group I, 10 had two WRKYGQK motifs and two $C_2H_2$-type zinc finger motifs (C-$X_{3\text{-}4}$-C-$X_{22\text{-}23}$-H-$X_1$-H), corresponding to two full WRKY domains. Although the protein encoded by *SbWRKY19* had only one WRKY domain, it belonged to group I on the phylogenetic tree. Fifty protein sequences with one WRKY domain and the $C_2H_2$-type zinc finger motif (C-$X_{4\text{-}5}$-C-$X_{23}$-H-$X_1$-H) were classified into group II. This group was further divided into five sub-groups, IIa, IIb, IIc, IId, and IIe, with 4, 8, 20, 6, and 12 members, respectively. Group III contained 31 members with one WRKY domain and the $C_2HC$-type zinc finger motif (C-$X_7$-C-$X_{23}$-H-$X_1$-C). *Sb*WRKY14 was unique in that it comprised of two WRKY domains with the $C_2HC$-type zinc finger motif (C-$X_7$-C-$X_{23}$-H-$X_1$-C). Thus, it had features associated with both group I and group III WRKYs, but was classified into group III based on its position in the phylogenetic tree. *Sb*WRKY6 and *Sb*WRKY43 did not belong to any group (Fig 2 and Table 1). Group II was the largest group and accounted for 53.2% of all putative *Sb*WRKYs, which is similar to reports in wheat, soybean (*Glycine max*), and pepper (*Capsicum annuum*). Overall, the classification of *Sb*WRKYs confirms their diversification, which suggests that different family members may have varied functions.

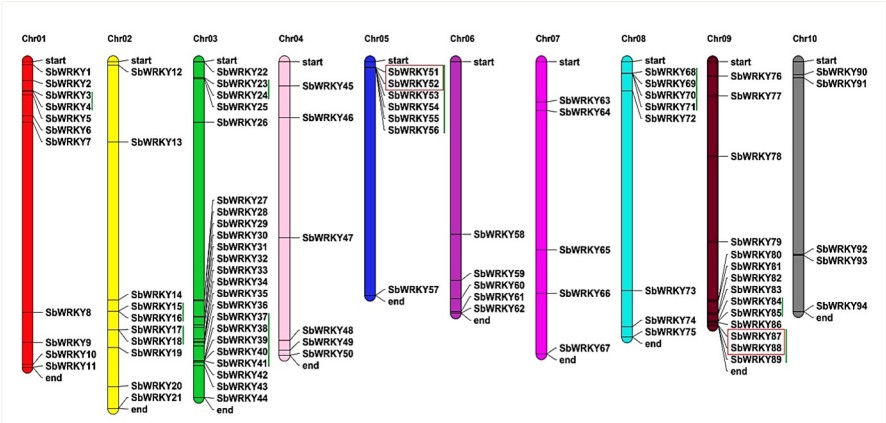

**Fig 1. Distribution of 94 *SbWRKY* genes on sorghum chromosomes.** Chr01-Chr10 above the colored bars indicates chromosome (Chr) numbers. The physical location of each *SbWRKY* gene is shown, and the gene name is indicated on the right side of each bar as *SbWRKY#*. Red boxes indicate tandem duplications, and green lines denote gene clusters.

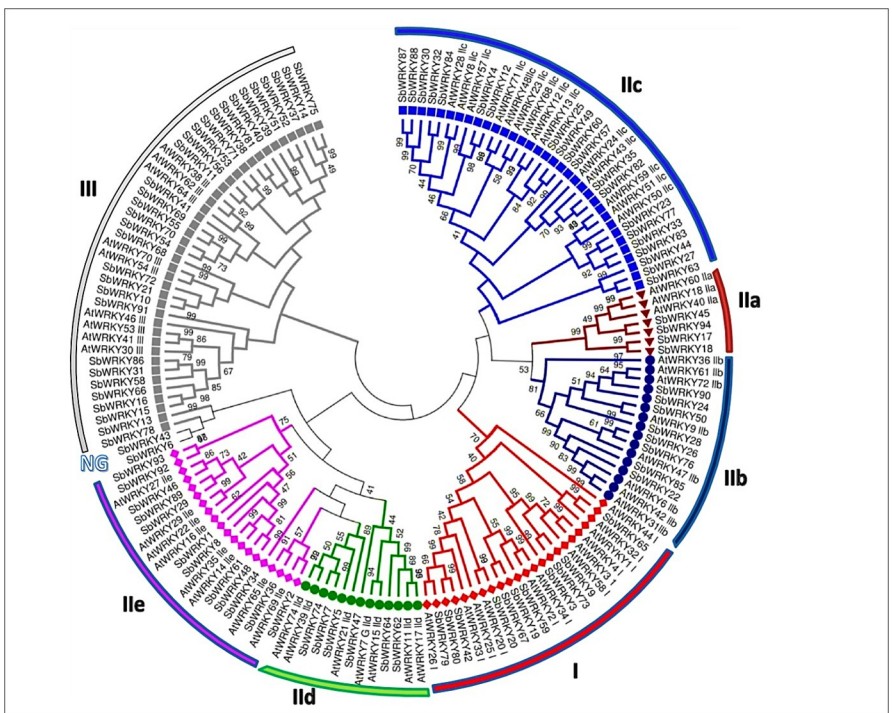

**Fig 2. Phylogenetic tree of WRKY members in sorghum and Arabidopsis.** *Sb*WRKY and *At*WRKY protein sequences were aligned with ClustalW, and a phylogenetic tree was constructed with MEGA7.0 using the neighbour-joining method and 1,000 bootstrap replicates. The members were divided into groups I, II, and III, and group II was further divided into subgroups IIa, IIb, IIc, Ild, and Ile.

The highly conserved heptapeptide motif WRKYGQK was present in 81 *Sb*WRKY proteins. We observed several heptapeptide variants in the remaining 13 proteins as follows: WRKY-GEK in six proteins (*Sb*WRKY11, *Sb*WRKY51, *Sb*WRKY52, *Sb*WRKY53, *Sb*WRKY71, and *Sb*WRKY81); WRKYGKK in five proteins (*Sb*WRKY23, *Sb*WRKY27, *Sb*WRKY33, *Sb*WRKY77, and *Sb*WRKY83); WRKYGSK in *Sb*WRKY14; and WRKSGQR in *Sb*WRKY75. Among the 94 identified *Sb*WRKYs, only 11 had two WRKY domains, whereas the remaining members had one WRKY domain. *Sb*WRKY protein sequences, genomic sequences, and CDS are provided in S2, S3, and S4 Files, respectively. *At*WRKY protein sequences are provided in S5 File.

## *SbWRKY* gene structure analysis

To obtain additional clues about the evolution of *SbWRKY* family members and their specific features, *SbWRKY* exon-intron structures were analyzed. The intron number of *SbWRKY* genes ranged from zero to seven, whereas their size varied. Among the 94 *SbWRKY* genes identified, 10 contained two exons (and one intron), 58 had three exons (two introns), eight had four exons (three introns), eight had five exons (four introns), five had six exons (five introns); the *SbWRKY14* gene had eight exons and seven introns (Fig 3). Four genes (*SbWRKY51*, *SbWRKY64*, *SbWRKY66*, and *SbWRKY88*) lacked introns.

The exon–intron distribution patterns showed some similarities in terms of their numbers and positions within the same group. However, there were also differences within groups. For instance, all genes in group II had zero to five introns, all *SbWRKY* genes in subgroup IIb had five introns except *SbWRKY22* and *SbWRKY50*, which had four, *SbWRKY24* had two, and

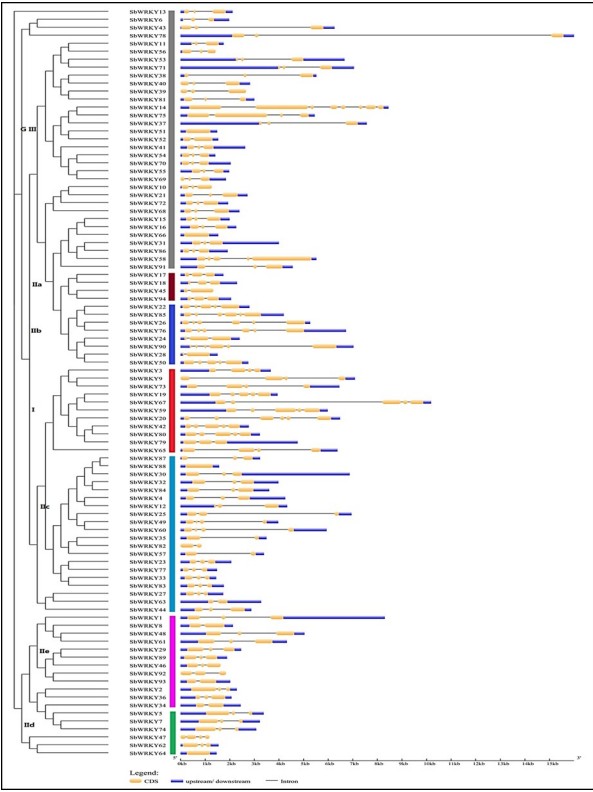

**Fig 3. Exon–intron structures of *SbWRKY* genes.** We used the Gene Structure Display Server (GSDS) for gene structure analysis and constructed the phylogenetic tree using MEGA v7.0. In the gene diagrams, blue bars indicate upstream and downstream UTRs, yellow bars indicate coding sequences (CDS), and black lines indicate introns.

*SbWRKY28* had one intron. Intron numbers in group III ranged from zero to seven, with *SbWRKY14* being the only gene with seven introns among all *SbWRKYs*. These results indicate that there is considerable structural variation among *SbWRKYs*, which may correspond to functional diversification between closely related members (Fig 3).

## Motif composition analysis of *Sb*WRKYs

MEME (version 4.11) was used to analyze all *Sb*WRKY protein sequences for conserved motifs, resulting in the identification of 20 distinct conserved motifs, ranging from 6 to 50 amino acids in length (Fig 4). Motifs 1, 2, 3, and 4 corresponded to the WRKY domain located at the C-terminus of sorghum WRKY proteins. Most *Sb*WRKY members within the same group or subgroup shared a similar motif composition. Motifs 12, 15, and 20 were unique to group III. Motifs 7, 10, 11, and 18 were unique to group IIb, and motif 14 was exclusively detected in group I. All group I members had two WRKY domains except for *Sb*WRKY19, as mentioned earlier, suggesting it may have lost its N-terminal WRKY domain. Motif 13 was unique to group IIe. Examples of motifs shared by different groups included motif 8, shared by groups IIe and IId, and motif 5, shared by groups I and IIc. Although *Sb*WRKY6 and *Sb*WRKY43 clustered with group III, they are not associated with any group; they contained motifs 1, 2, 6, 3, and 17 (Fig 4). Groups IIe and IId were two close subgroups in the phylogenetic tree, and the vast majority of their members contained motifs 8, 2, 4, 3, and 17. Both subgroups had a similar domain arrangement, which may be indicative of functional similarity.

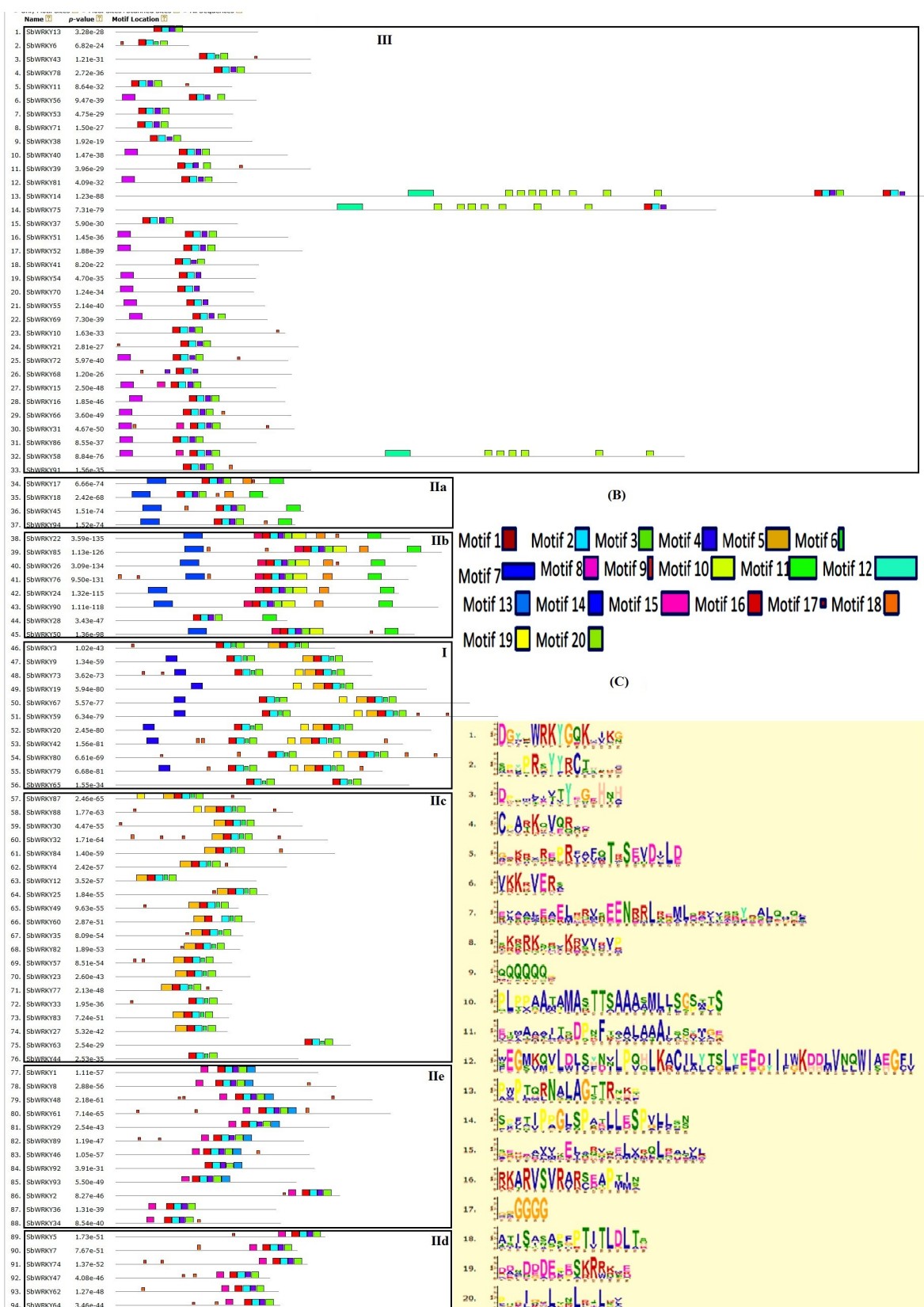

**Fig 4. Motif analysis of *Sb*WRKYs.** (a) The distribution of 20 conserved motifs identified by MEME in the different groups of *Sb*WRKYs. (b) Each motif is indicated by a different color. (c) Sequence logos for motifs 1–20.

Some motifs occurred only in a few *Sb*WRKY members, such as motifs 12 and 20, which were only present in *Sb*WRKY14, *Sb*WRKY75, and *Sb*WRKY58.

## Gene ontology annotation and analysis of *cis*-acting elements

Gene ontology (GO) annotations of 94 *Sb*WRKY proteins were analyzed using the Blast2GO tool. The *SbWRKY* target genes were categorized into different functional groups under three main categories including biological processes, molecular functions, and cellular components (Fig 5). Under the larger umbrella of biological processes, most *SbWRKYs* were identified as being involved in the regulation of cellular processes, biosynthetic processes and different metabolic processes, as well as response to different stimuli, signalling, cell communication, and responses to other organisms, chemicals and stress. The molecular functions of *SbWRKYs* were associated mostly with DNA-binding, DNA-binding transcription factor activity, catalytic activity, acting on a protein, and hydrolase activity. The cellular component of this protein family included organelle and intracellular organelle (Fig 5). In addition, all *Sb*WRKY proteins were predicted to be localized in the nucleus.

*Cis*-acting elements within promoters are the binding sites through which transcriptional regulation is enacted. We therefore, extracted the 1.5-kbp promoter regions upstream of all *SbWRKY* genes from the sorghum genome assembly to identify *cis*-acting elements using the online tool PlantCARE. Thus, various *cis*-acting regulatory elements were found in all *SbWRKY* genes promoter regions. Featuring prominently in our list of *cis*-elements were stress-responsive elements, including MBS (MYB transcription factor binding site involved in drought inducibility), LTR (low-temperature responsive element), ARE (anaerobic induction responsive element), TC-rich repeats (defense-responsive and stress-responsive elements, WUN-motif (wound-responsive elements), and GC-motif (anoxic specific inducibility element). Phytohormone-responsive elements: ABA-responsive element (ABRE), methyl jasmonate (MeJA) responsive element (TGACG-motif and CGTCA-motif), auxin-responsive elements (AuxRR-core and TGA-element), salicylic acid-responsive element (TCA-element), and gibberellin-responsive element (GARE-motif). Multiple light-responsive elements were

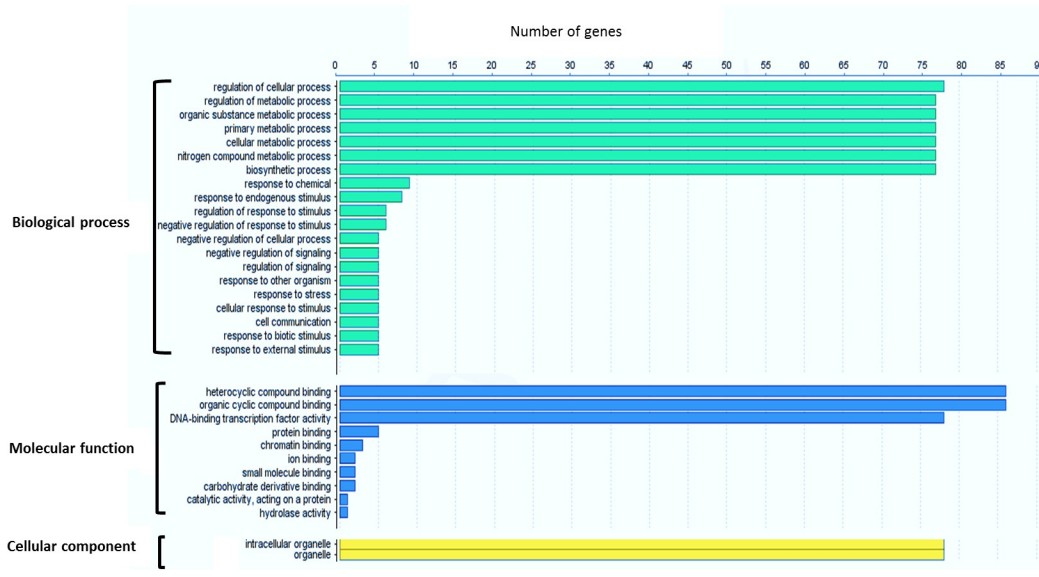

**Fig 5. Gene ontology analysis of identified *Sb*WRKYs.** The enrichment analysis shows the involvement of *Sb*WRKY in biological processes, molecular functions, and cellular components.

present in the promoters of *SbWRKY* genes, including Sp1, TCT-motif, GT1-motif, GATA-motif, GA-motif, BoX II, and G-box, as well as elements associated with development, including the CAT-box (element related to meristem expression), and the o2-site (metabolism regulation). The promoter related and binding sites elements were found included TATA-box, CAAT-box, A-box, HD-Zip, and W-box (a classic WRKY DNA-binding motif). Many unknown functions were detected included AAGAA-motif, DRE core, and MYB. The most common *cis*-acting regulatory elements in the *SbWRKY* promoter regions were TGACG-motif, ABRE, CGTCA-motif, CAAT-motif, MYB, TATA-box, and G-box. We identified W-box elements in the promoters of 38 *SbWRKY* genes. All *SbWRKY* genes contained at least one stress-responsive element along with other *cis*-elements, reflecting their potential functional variation (S6 File).

## Digital expression analysis of *SbWRKY* genes at different growth stages and in different tissues

As a preliminary survey of the potential roles of *SbWRKY* genes during sorghum growth and development, the temporal and spatial expression profiles of *SbWRKY* genes were investigated using microarray data available from SorghumFDB. We used the Genevestigator platform for the analysis, and present the results as heatmaps. The sorghum accessions represented in the microarray datasets included R159, Fremont, Atlas, PI455230, PI152611, and AR2400 [41]. The microarray datasets comprised 37 samples representing leaves, roots, shoots, stems (pith and rind), and internodes. Seventy-eight of the 94 *SbWRKY* genes were represented in the microarray data and displayed distinct expression patterns across all tested tissues. For example, all 78 genes were expressed in leaves, 76 in roots, 77 in shoots and pith, 76 in internodes, and 75 in rind. Thirty-two of the 78 genes exhibited high expression levels in at least one tissue (Fig 6A). The number of genes with high expression levels (>65% expression) varied between tissues, although roots showed the highest number of highly expressed *SbWRKY* genes with 17 members, followed by nine in pith, seven in leaves, six in rind, five in internodes and four in shoots. The most highly expressed *SbWRKY* genes were *SbWRKY19*, *SbWRKY83*, *SbWRKY45*, *SbWRKY79*, *SbWRKY5*, *SbWRKY42*, *SbWRKY73*, *SbWRKY22*, *SbWRKY34*, *SbWRKY72*, *SbWRKY25*, and *SbWRKY70* in different tissues (Fig 6A). Notably, *SbWRKY72* was highly expressed in all tissues. By contrast, *SbWRKY70* expression was only detected in leaves and shoot. Clustering analysis of *SbWRKY* expression patterns grouped rind and pith, this is consistent with their biological features.

*SbWRKY* expression patterns at different growth and developmental stages (seedling, stem elongation, flowering, boot, and dough stage) were also analyzed (Fig 6B). *SbWRKY* genes were expressed differently (up- or down-regulated) at all stages, and those with high expression at different stages included *SbWRKY74*, *SbWRKY75*, *SbWRKY19*, *SbWRKY5*, *SbWRKY45*, *SbWRKY79*, *SbWRKY25*, and *SbWRKY72*. *SbWRKY* expression was slightly higher during the seedling, flowering, and dough stages, suggesting that they may be involved in stress responses during sensitive developmental stages to improve plant tolerance (Fig 6B).

Hierarchical clustering analysis of the expression patterns of 78 *SbWRKY* genes under different environmental stresses was performed in Genevestigator. Two major clusters were obtained, which divided *SbWRKY* genes into two groups. The first major cluster consisted of highly expressed genes, including *SbWRKY45*, *SbWRKY79*, *SbWRKY83*, and *SbWRKY16*, under various abiotic stress conditions such as drought, salt and ABA (Fig 6C). The second major cluster contained several sub-clusters of *SbWRKY* genes with different expression patterns, i.e., up- or downregulated at least 2.5-fold (in absolute terms), in response to drought, salt, ABA in different sorghum tissues (Fig 6C). Several genes were found to have a stable

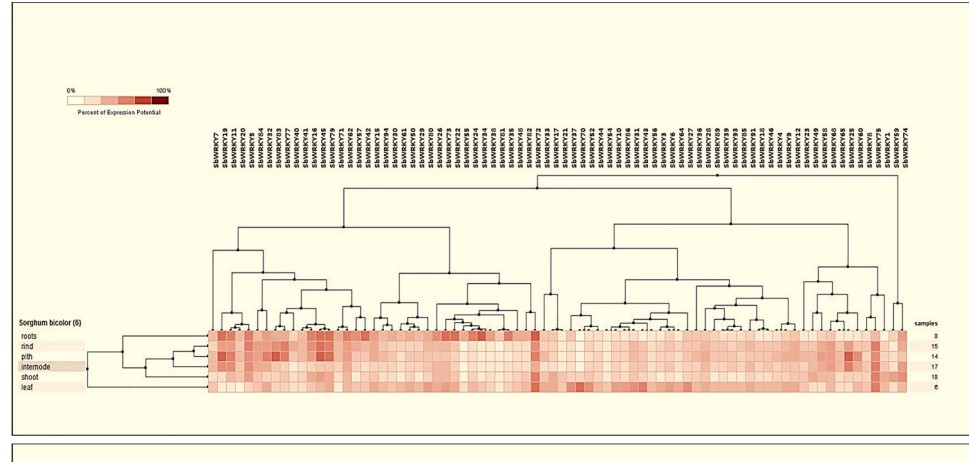

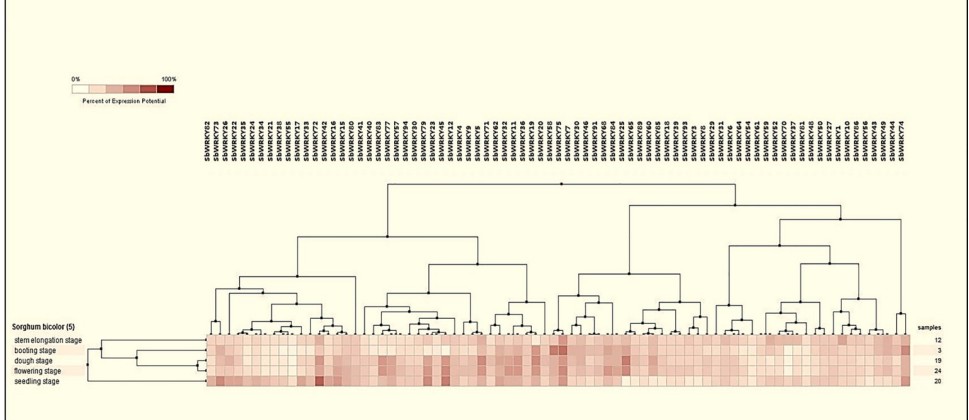

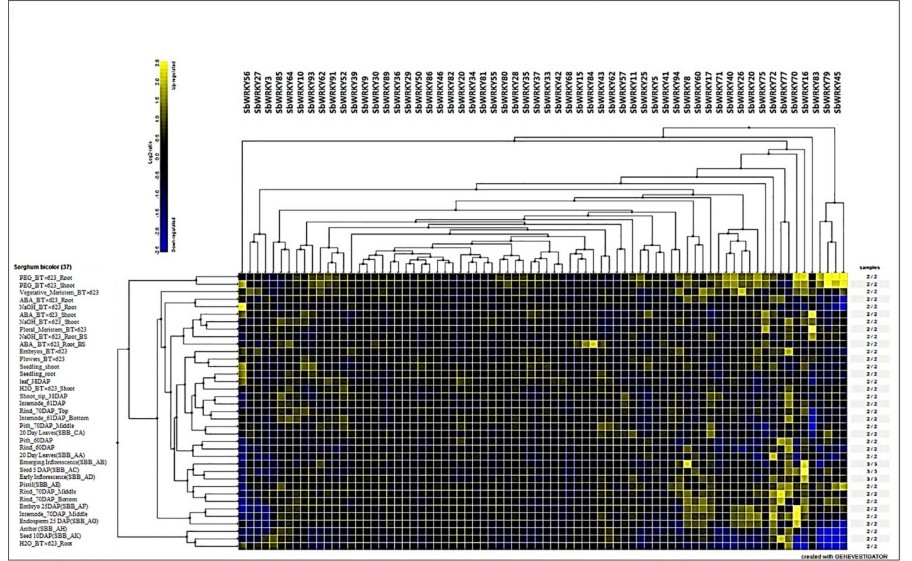

**Fig 6. Heatmaps of *SbWRKY* gene expression.** *SbWRKY* expression levels in different tissues (a) and at different growth stages (b). (c) Hierarchical clustering of *SWRKY* gene expression patterns under different environmental conditions, including drought, ABA, heat, salt and combination stress.

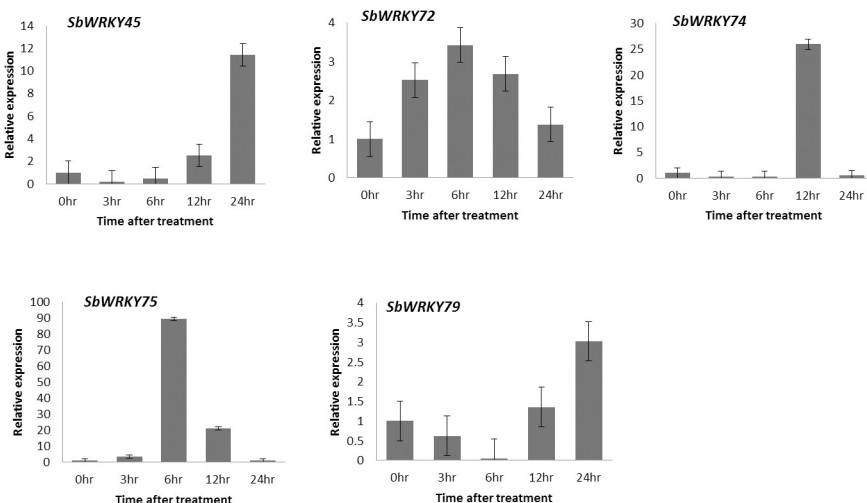

**Fig 7. Relative expression of selected *SbWRKY* genes in response to drought stress.** Relative expression levels of five *SbWRKY* genes under drought stress. Genes expression was analyzed by RT-qPCR, with the 0 h sample used as untreated control (expression = 1). Error bars represent standard errors; data were calculated using the $2^{-\Delta\Delta Ct}$ method.

expression level across different tissues and may therefore be considered constitutively expressed. We hypothesize that other *SbWRKY* genes with low expression levels may work cooperatively with other proteins throughout plant development.

## qRT-PCR expression analysis of *SbWRKY* genes in response to drought stress

Previous studies proved that *WRKY* genes involved in plant responses to drought stress in several crops such as maize, wheat, and rice [42]. To investigate the role of *SbWRKY* genes in drought responses in sorghum, we selected five genes (*SbWRKY45*, *SbWRKY72*, *SbWRKY74*, *SbWRKY75*, and *SbWRKY79*) for expression analysis by qRT-PCR in the shoot of sorghum seedlings subjected to drought stress. qRT-PCR results revealed that transcripts levels for these five *SbWRKY* genes were remarkably increased under drought stress at different time points (Fig 7) suggesting that these genes may function in this process. The relative expression of our selected *SbWRKY* genes peaked at different time points. *SbWRKY75* exhibited the highest expression level, with an 89-fold change in expression after 6 h of drought stress, with a final decline after 24 h. The peak expression of *SbWRKY74* occurred 12 h after the onset of drought stress, whereas *SbWRKY45* and *SbWRKY79* expression peaked 24 h into drought stress (Fig 7). Although the peak expression of *SbWRKY72* and *SbWRKY75* occurred at 6 h, the expression of *SbWRKY72* gradually increased to this level, then decreased gradually at later time points. *SbWRKY79* showed the least induction during drought stress relative to the other genes tested here. Overall, the expression pattern of these selected *SbWRKY* genes under drought stress conditions suggests that different *SbWRKY* genes may play an essential role in drought stress tolerance.

## Discussion

WRKY TFs are key regulators of many processes in plants, including responses to abiotic and biotic stresses. In both model and non-model plants, considerable progress has been made towards identifying and functionally characterizing WRKY TFs, and many *WRKY* genes have been found to promote stress tolerance [42]. The completed genome assembly of sorghum

now makes it possible to perform a genome-wide analysis of the *SbWRKY* gene family. We identified a total of 94 *SbWRKY* genes in *S. bicolor*, which is slightly higher than in other species; for example, there are 75 *WRKY* genes in *Arabidopsis* [1], 71 in sesame [25], 79 in potato, 85 in cassava (*Manihot essculenta*) [43], and 59 in grapevine [44]. In contrast, sorghum contains fewer *WRKY* genes than maize, soybean, or rice [23, 45, 46]. The present findings in sorghum, an important cereal crop and a model plant for drought tolerance, add to the recent identification of *WRKY* genes in a variety of plant species, including chickpea [26], Chinese jujube (*Ziziphus jujube*) [47], sugar beet (*Beta vulgaris*) [48], coffee (*Coffea arabica*) [49], pepper [50], eggplant (*Solanum melongena*) [51], Asian legume crops [52], sweet potato (*Ipomoea batatas*) [53], and pearl millet (*Pennisetum glaucum*) [54].

*SbWRKY* genes are distributed on all 10 sorghum chromosomes. Previous studies mapped many traits, such as stay-green phenotypes, lodging tolerance, pre-flowering drought tolerance, and yield-related components, to various chromosomal locations in sorghum [55, 56, 57]. Several *SbWRKY* genes map within these candidate regions; for example, *SbWRKY26* is located on chromosome 3, close to a mapping interval for stay-green and pre-flowering drought tolerance Quantitative Trait Loci (QTL) in sorghum. Likewise, *SbWRKY67* maps to chromosome 7, in a genomic region associated with a lodging tolerance QTL, whereas *SbWRKY59* maps to the same region associated with pre-flowering drought tolerance, flowering time, and stay-green QTLs. These results suggest that several *SbWRKY* genes might control or contribute to stay-green, pre-flowering drought tolerance, or other traits related to stresses tolerance.

Gene duplication events affect genome expansion, family size, and the distribution of genes on chromosomes. Distinct types of duplication events, such as tandem and segmental duplication, differ in terms of the resulting number of gene copies and their distribution, and these factors are important for functional prediction. Tandem duplication of chromosome regions can also give rise to a cluster of family members, with subsequent structural and functional divergence over time leading to the expansion and evolution of the gene family [58]. Our analysis identified two tandem duplications of *SbWRKY* genes, which is a smaller number than that reported for potato [27], rice [24], wheat [22], or Arabidopsis [59]. Despite these differences, tandem duplications may have shaped the evolution of the *SbWRKY* gene family in sorghum.

Both tandem duplications and segmental duplications have played essential roles in the evolution and diversification of the *WRKY* gene family in plant species [60]. These evolutionary events were assigned to three types of gene duplication: tandem, segmental and whole-genome duplications [61]. Duplication events are significant for *WRKY* diversification, as duplicated *WRKY* genes may acquire new functions. In this study, we focused on tandem duplication events. We hypothesize that tandem duplications have played an important role in the evolution and diversification of *WRKY* genes in sorghum, although the expansion of this gene family likely arose mainly through other events. Tandem duplications were a critical but recent gene duplication contributor in *Arabidopsis* [62]. Previous studies have shown that gene duplication greatly accounts for new genes [63]. Gene duplication may result in sub-functionalization: for example, expansion of the functions among the wheat *WRKY* gene family members has occurred through tandem duplication and whole-genome duplication [64]. Tandem duplications and gene clusters have been previously described for multiple *WRKY* genes in rice within the same intergenic region [65].

The presence of the conserved WRKY domain, which binds to the W-box motif in the promoters of WRKY target genes, is the most essential characteristic of the WRKY family [66, 67]. We performed a classification of *SbWRKY* genes here according to the approach used in other crop species, based on phylogenetic tree topology. We also adopted the divisions of WRKY family members into the same groups and subgroups described in Arabidopsis: groups I, II, and III according to the number of WRKY domains and the type of zinc finger motif, along

with further subdivision of group II into subgroups IIa, IIb, IIc, IId, and IIe [1]. In this study, 11, 50, and 31 *SbWRKY* genes were classified into groups I, II, and III, respectively; two genes did not belong to any group. Group II was the largest with 50 members and accounted for 53.2% of all *SbWRKY* genes. These results are consistent with *WRKY* group sizes in sesame, Arabidopsis, and sugar beet [25, 1, 46]. Among the subgroups, subgroup IIc was the largest with 20 *SbWRKY* genes, or 40% of the genes assigned to group II (Fig 2), which is similar to results in sugar beet [48], *Arabidopsis* [1], and soybean [68].

Whereas most *Sb*WRKY proteins contained the conserved WRKYGQK motif, other hepta-peptide variants were identified in 13 *Sb*WRKY proteins (*Sb*WRKY11, *Sb*WRKY14, *Sb*WRKY23, *Sb*WRKY27, *Sb*WRKY33, *Sb*WRKY51, *Sb*WRKY52, *Sb*WRKY53, *Sb*WRKY71, *Sb*WRKY75, *Sb*WRKY77, *Sb*WRKY81, and *Sb*WRKY83; Table 1). Similar variations in the heptapeptide motif were also identified in potato [27]. Several examples indicate that these differences may affect the binding ability of WRKY TFs to the W-box element. For example, two soybean WRKYs with the WRKYGKK motif variant were unable to bind to the W-box element [66]. In tobacco (*Nicotiana tabacum*), *Nt*WRKY12 with the WRKYGKK motif bound to TTTTCCAC instead of the W-box consensus sequence (TTGACT/C) [69]. Therefore, further investigation is warranted to identify the preferred DNA-binding sequences associated with different WRKYGQK-like motifs.

Group-specific patterns were detected in their exon–intron structure. Indeed, *SbWRKY* genes within the same group had similar exon–intron patterns. The number of introns in *SbWRKY* genes ranged from 0 to 7, which is similar to that reported in chickpea [26]. Four *SbWRKY* genes lacked an intron, indicating that intron loss may have occurred during the evolution of this gene family; a similar observation was reported in the rice *WRKY* gene family [46]. Intron-less genes within plant gene families may imply their close relationship. Intron-less genes is not a specific feature of the *SbWRKY* gene family, as they have been reported in other gene families like GRAS-domain TFs, F-box TFs [70], small auxin-up RNAs [71], and DEAD-box RNA helicases [72]. Intron-less genes may arise from one of three major mechanisms: retroposition (the integration of a sequence derived from RNA into the genome), duplication of existing intron-less genes, and horizontal gene transfer [73].

The variations in intron sizes within and between *SbWRKY* groups may have resulted from duplication, inversion, and/or fusion events [74]. These results were similar to findings in wheat [22], carrot (*Daucus carota*) [74], and cassava [43]. Overall, the diversification of the exon-intron pattern will provide important clues about the evolution of *SbWRKY* genes.

Among 20 identified functional motifs in *Sb*WRKY proteins, motifs 1, 2, 3, and 4 corresponded to WRKY domains containing zinc finger domains that are present in most *Sb*WRKY members. Motif 8 represented the nuclear localization signal (NLS), mainly distributed in subgroups IId and IIe. As described previously, members of subgroup IId possess an NLS and a conserved calmodulin-binding domain. Interestingly, three members of subgroup IId (*Sb*WRKY74, *Sb*WRKY62, and *Sb*WRKY64) contained the conserved HARF motif (RTGHARFRR [A/G] P), which was also identified in Arabidopsis and poplar (*Populus trichocarpa*) WRKY subgroup IId, although the function of this motif is unknown [75, 1]. Some motifs were located nearby the WRKY domain, for instance, motifs 6, 9, and 17. Just as phylogenetic analysis divided *Sb*WRKY genes into groups I-III and subgroups IIa-e, the presence or absence of shared motifs between *Sb*WRKY proteins followed the same general separation into groups, consistent with previous studies [76–79]. Indeed, each of these motifs occurs in most subgroups, and each subgroup can be distinguished based on the motifs they present. Group-specific motifs might be involved in responses to a given biological process [80] and may provide clues about their potential function. The functions of other motifs identified by MEME are yet to be elucidated.

The promoter regions of 94 *SbWRKY* genes exhibited various conserved *cis*-acting regulatory elements involved in various functions, such as abiotic and biotic stress responses (MBS, LTR, ARE, TC-rich repeat, and GC-motif) and phytohormone regulation (ABRE, TCA-element, TGA-element, TGACG-motif, CGTCA-element, AuxRR-core, and GARE-motif). The presence of many *cis*-acting elements mediating responses to environmental stress and phytohormones indicates their involvement in different biological processes. Regulation of *WRKY* expression may occur via binding of a WRKY TF to W-box or by the binding of another TF to a different *cis*-element along *WRKY* promoters [81]. The wheat *Ta*WRKY2 and *Ta*WRKY19 exert their regulation of gene expression by binding to the promoter regions of target genes when overexpressed in Arabidopsis [82]. WRKY TFs may regulate the expression of their encoding genes by binding to their promoter or may regulate other WRKY TFs by cross-regulation [83]. Consistent with this hypothesis, the promoters of 38 *SbWRKY* genes had one or more W-boxes.

The essential roles of WRKY TFs in plant growth, development, and stress tolerance are supported by *WRKY* gene expression data from several species. From extensive studies in the model plant *Arabidopsis*, many *AtWRKYs* have been functionally characterized. Therefore, identifying the closest *Arabidopsis* homologue(s) of individual *SbWRKYs* may provide a hint as to their potential functions. For example, the highly expressed *SbWRKY72* gene is most closely related to the *Arabidopsis AtWRKY70* and *AtWRKY54*, which have been reported to modulate osmotic stress tolerance by regulating stomatal aperture [84].

*SbWRKY79*, *SbWRKY80*, and *SbWRKY42* are putative orthologs of *AtWRKY25*, *AtWRKY26*, and *AtWRKY33*, which regulate heat shock proteins and the heat-induced ethylene-dependent response [85]. *AtWRKY53* and *AtWRKY70* belong to group III and both play important roles in leaf senescence [86]. *SbWRKY72* and *SbWRKY75* also belong to group III and are highly expressed in leaves. In addition, *AtWRKY70* plays a critical role in osmotic stress signalling and plant defense responses in *Arabidopsis* [87]. *SbWRKY19* and *SbWRKY73*, members of group I, are expressed in root tissues, although their putative *Arabidopsis* counterpart *AtWRKY34* is involved in regulating gene expression during tapetum formation [88]. Several group II-a *WRKY* genes have documented roles in biotic stress responses in *Arabidopsis* [42]. Moreover, *AtWRKY8*, *AtWRKY50* and *AtWRKY57* are involved in phytohormones signalling and pathogen responses [89]. The highly expressed sorghum gene *SbWRKY45* is the putative ortholog of *AtWRKY18*, *AtWRKY40*, and *AtWRKY60*, which are involved in abscisic acid signalling and abiotic stress [90]. Additionally, *SbWRKY45* is the putative ortholog of maize *ZmWRKY40*, which confers drought resistance when expressed in transgenic *Arabidopsis* [12]. A putative ortholog of *SbWRKY8* is Arabidopsis *WRKY57*, which can improve drought tolerance through elevated abscisic acid levels [52].

Based on the available sorghum transcriptomic data, the present analysis of *SbWRKY* genes revealed their different expression patterns at different growth stages and in different tissues. As indicated by the above examples, the known functions of putative Arabidopsis *WRKY* orthologs, together with expression data, will guide future functional analyses of *SbWRKY* genes, with a focus on their roles in responses to environmental stress. Highly expressed *SbWRKY* genes identified during our digital expression analysis of published sorghum microarrays were validated by qRT-PCR, which also provided general clues on *SbWRKY* responses to drought stress. We tested five genes that belonged to group I (*SbWRKY79*), IIa (*SbWRKY45*), IId (*SbWRKY74*), and III (*SbWRKY72* and *SbWRKY75*). All selected genes were upregulated in response to drought. *WRKY* genes from the same groups were reported to play important roles in other plants. For example, the expression of sweet potato group I members *ItfWRKY66*, *ItfWRKY69*, and *ItfWRKY80* was induced in response to drought, cold and salt stresses [53]. Furthermore, *AtWRKY25* overexpression improved heat and salt stress tolerance in *Arabidopsis*

[91]. Its putative ortholog *SbWRKY79* was induced by drought as shown by RT-qPCR. Several putative *SbWRKYs* orthologs have been confirmed to be involved in drought stress tolerance in other crops. For example, *SbWRKY45* was orthologous to maize *ZmWRKY40*, which is itself involved in drought stress tolerance [12].

Since we only tested a small fraction of *SbWRKY* genes, other *SbWRKY* genes may also be involved in drought stress responses. Therefore, further investigation into *SbWRKY* expression under other abiotic stress conditions (cold, salinity, and heat) is necessary. Our results provide several promising *SbWRKY* candidates for these future studies.

## Conclusion

This study identified 94 *WRKY* genes in sorghum, and the following analyses were performed: characterization and classification, gene structure analysis, chromosome mapping, and conserved motif analysis. *SbWRKY* gene expression profiles indicated that *SbWRKY* genes may be important in different tissues and at different developmental stages. Several *SbWRKY* genes displayed tissue-specific expression. Besides, several *SbWRKY* genes were highly expressed in response to environmental stresses. qRT-PCR analysis revealed several *SbWRKY* genes induced by drought stress. In Arabidopsis, many *AtWRKYs* regulate abiotic and biotic stress responses, and the available information about specific *WRKY* members will facilitate functional validation and characterization of their putative orthologs in sorghum. Overall, the present findings provide a foundation for future functional analyses of *SbWRKY* genes in response to abiotic and biotic stress in sorghum.

## Supporting information

**S1 File. Detailed information on sequence similarity of putative paralogous pairs for tandem duplications.**
(DOCX)

**S2 File. *Sb*WRKY protein sequences.**
(DOCX)

**S3 File. *SbWRKY* genomic sequences.**
(DOCX)

**S4 File. *SbWRKY* CDS sequences.**
(DOCX)

**S5 File. *At*WRKY protein sequences.**
(DOCX)

**S6 File. *Cis*-acting elements in *SbWRKY* genes.**
(XLSX)

**S7 File. *SbWRKY* specific primers used for RT-qPCR.**
(DOCX)

## Acknowledgments

The authors sincerely thank Professor Fuyao Zhang, Sorghum Institute, Shanxi Academy of Agriculture Sciences, Shanxi, for providing sorghum seeds used in this research. We thank Anfal for helpful suggestions and advice.

## Author Contributions

**Conceptualization:** Elamin Hafiz Baillo.

**Data curation:** Elamin Hafiz Baillo.

**Formal analysis:** Muhammad Sajid Hanif.

**Supervision:** Zhengbin Zhang.

**Validation:** Elamin Hafiz Baillo.

**Visualization:** Elamin Hafiz Baillo, Yinghui Guo, Ping Xu.

**Writing – original draft:** Elamin Hafiz Baillo.

**Writing – review & editing:** Elamin Hafiz Baillo, Yinghui Guo, Soad Ali Algam.

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
