## [Decision Letter · Decision Letter 0]

9 Apr 2020

PONE-D-20-02381

Genome-wide identification of WRKY transcription factor family members in sorghum (Sorghum bicolor (L.) moench)

PLOS ONE

Dear Dr Baillo,

Thank you for submitting your manuscript to PLOS ONE. After careful consideration, we feel that it has merit but does not fully meet PLOS ONE’s publication criteria as it currently stands. Therefore, we invite you to submit a revised version of the manuscript that addresses the points raised during the review process.

We would appreciate receiving your revised manuscript by May 24 2020 11:59PM. To enhance the reproducibility of your results, we recommend that if applicable you deposit your laboratory protocols in protocols.io, where a protocol can be assigned its own identifier (DOI) such that it can be cited independently in the future. For instructions see: http://journals.plos.org/plosone/s/submission-guidelines#loc-laboratory-protocols

We look forward to receiving your revised manuscript.

Kind regards,

Ramasamy Perumal, Ph.D.

Academic Editor

PLOS ONE

Additional Editor Comments (if provided):

Authors are requested to make major revision as suggested by Reviewer 2: Improve the quality of all figures, succinct presentation of introduction part, inclusion of gene ontology and Cis-elements analyses for the WRKY genes, clear explanation gene duplication event, provide more detailed discussion with more citation for clear justification of the results, identify the function of each motif identified by MEME and discuss about these functions, and how these motifs interact with the WRKY motif with clear presentation of the manuscript following the standard grammar for clear understanding.

Authors are also requested to make necessary changes suggested by reviewers 1 and 2 in the revised version.

"This project was supported by the Innovation 471 Academy for Seed Design, Chinese Academy of Sciences. The authors sincerely thank the Center for Agricultural Resources Research, Institute of Genetics and Developmental Biology (CAS) for providing the facilities."

"The authors received no specific funding for this work "

4. Please include a copy of Table 2 which you refer to in your text on page 14.

Reviewers' comments:

Reviewer's Responses to Questions

**Comments to the Author**

1. Is the manuscript technically sound, and do the data support the conclusions?

Reviewer #1: Yes

Reviewer #2: Partly

Reviewer #3: Yes

2. Has the statistical analysis been performed appropriately and rigorously? 

Reviewer #1: Yes

Reviewer #2: No

Reviewer #3: N/A

3. Have the authors made all data underlying the findings in their manuscript fully available?

Reviewer #1: Yes

Reviewer #2: No

Reviewer #3: Yes

4. Is the manuscript presented in an intelligible fashion and written in standard English?

Reviewer #1: Yes

Reviewer #2: No

Reviewer #3: Yes

5. Review Comments to the Author

Reviewer #1: The paper is generally well written in good english, but a bit of editing is required; I have attached an edited pdf with suggestions

The only thing I feel needs to be addressed is the situation of WRKY genes with no introns. Could these be processed pseudogenes, ie reverse transcribed from a mRNA and reincorporated into the genome. did they check for functional promoter signals, expression etc. The issue should be considered and addressed.

Reviewer #2: Review of manuscript PONE-D-20-02381

Title: “Genome-wide identification of WRKY transcription factor family members in sorghum (Sorghum bicolor (L.) moench)”

Major Comments:

1. The poor quality of the figures makes difficult the review of this manuscript.

2. Please rewrite the introduction, it’s longer that discussion section, some information from the introduction can be used in the discussion part.

3. How the authors determined to use PEG at 20% for drought stress treatment?

4. What the authors refer to incomplete domain, and why these genes were not included in the analysis, please justify.

5. Please include the Gene Ontology analysis for the WRKY genes.

6. It would be very useful if the authors provide more information about the expected subcellular location based on the subfamilies of WRKY genes.

7. Please include the Cis-elements analysis for the WRKY genes.

8. Please include the parameters that authors used to identify and classify gene duplication models.

9. Please explain which Gene duplication event (tandem or segmental) contributed predominantly to the expansion of the WRKY genes and why.

10. Author mention in line 154-161 that they survey the SbWRKY expression profiles under different stress, but the results are not included.

11. Authors should discuss more deeply all their results, for example, authors can discuss about the changes in the expression and if these changes can affect the plant development, or why one group of genes are more expressed in determinate stage, etc.

12. Include more references through all the manuscript specifically in the discussion to increase the value of the paper.

13. Please identify the function of each motif identified by MEME and discuss about these functions, and how these motifs interact with the WRKY motif?

Minor Comments:

1. Authors should have one or more native English speakers review the manuscript before resubmission.

2. In references and manuscript, please look carefully the gene and species names. They should be in italics.

3. Please check grammar throughout the manuscript.

Reviewer #3: Manuscript summary:

WRKY transcription factors are shown to confer protection against biotic and abiotic stressors in model plants. However, limited knowledge exists on this family of transcription factors in sorghum. Baillo et al. studied WRKY transcription factors in the sorghum genome to enable their utilization in improving sorghum stress tolerance. Genome-wide analysis identified the presence of 94 WRKY transcription factors with considerable structural variation. Based on conserved domains and motifs in Arabidopsis, SbWRKY family transcription factors were categorized into three groups. The authors identified WRKY were distributed on all 10 chromosomes and some were present in gene clusters. Publicly available microarray data from previous sorghum studies was used to identify specific WRKY gene expression under different environments. In this study, evaluating the expression of five WRKY genes under drought stress in seedlings showed differential regulation under drought. Knowledge of WRKY transcription factors generated from this manuscript will be immensely helpful in their future characterization under different biotic and abiotic stressors.

The authors have conducted thorough analyses of WRKY transcription factors in sorghum. The manuscript is well written and very informative. This manuscript has similarities with recently published reports on maize, rice, chickpea, sugar beet, and soybean WRKY transcription factors in PLOS One, BMC Plant Biology, and Nature Communications. Based on increasing interest in utilization of transcription factors in crop improvement, this manuscript will be very helpful for characterizing WRKY genes in sorghum.

Major suggestions:

Figure 5C. Very good heat map representing WRKY expression under different environments. Please include what each environment represents in the figure legend to help readers understand gene expression under a specific stress environment.

Table 1: Include the percentage of protein similarity between different WRKY genes with reference to previously characterized SbWRKY conferring seed shattering phenotype. This information will help readers understand structural variation between WRKY proteins.

Line338: What is the rationale for selecting these five wrky genes for qPCR in your drought tolerance experiment? Include it in the M&M or discussion section.

Figure2: In addition to phylogenetic analysis of sorghum and Arabidopsis WRKY genes, include syteny analysis of sorghum and Arabidopsis WRKY genes. This information will help on studying sorghum orthologs of previously characterized WRKY genes in model plants.

Line91: Include gene ID of SpWRKY.

Minor suggestions:

Lines 118 and 195: Change Morocoshi to Morokoshi

Line 375: Missing reference

Abbreviate the gene names when used for the first time. SPL1 sweet potato and GAPDH

Table 1: WRKY88, include Sobic ID for this gene.

6. PLOS authors have the option to publish the peer review history of their article (what does this mean?). If published, this will include your full peer review and any attached files.

Reviewer #1: No

Reviewer #2: No

Reviewer #3: Yes: Sandeep R Marla

---

## [Author Response · Author response to Decision Letter 0]

18 May 2020

Response to Reviewers:

Reviewer 1 Comments

We thank the reviewer for their thorough review and believe their input has been invaluable in making our manuscript more balanced.

Point 1: the only thing I feel needs to be addressed is the situation of WRKY genes with no introns. Could these be processed pseudogenes, ie reverse transcribed from a mRNA and reincorporated into the genome. Did they check for functional promoter signals, expression etc. The issue should be considered and addressed

Response: A gene may lack an intron due to one of three potential mechanisms, one of which is retroposition, as stated by the reviewer. The lack of introns in several WRKY genes may indicate their close evolutionary relationship. We identified four WRKY genes in sorghum with no introns. Two of these genes were found in tandem duplication. Such intronless genes have been identified in other gene families in other species, as discussed in the Discussion. 

Reviewer 2 Comments

We would like to thank the reviewer for their thorough review and efforts towards improving our manuscript. 

Point 1: The poor quality of the figures makes difficult the review of this manuscript.

Response: We have now improved the quality of all figures. We assessed all new Figures via the PACE tool provided by PLOS.

Point 2: Please rewrite the introduction, it’s longer than discussion section, some information from the introduction can be used in the discussion part.

Response: We have shortened the introduction. 

Point 3: How the authors determined to use PEG at 20% for drought stress treatment?

Response: We have established these conditions based on the previous work, now cited in Materials and Methods. 

Point 4: What the authors refer to incomplete domain, and why these genes were not included in the analysis, please justify. 

Response: All members of group I are classified as having two WRKY domains; however, SbWRKY29 only has one WRKY domain, although it appears to belong to group I based on the phylogenetic analysis. We now make this point clear in the Results section and again in the Discussion. 

Point 5: Please include the Gene Ontology analysis for the WRKY genes. 

Response: Done.

Point 6: It would be very useful if the authors provide more information about the expected subcellular location based on the subfamilies of WRKY genes.

Response: Gene Ontology analysis includes predicted subcellular localization included in our revised manuscript. 

Point 7: Please include the Cis-elements analysis for the WRKY genes.

Response: Done. 

Point 8: Please include the parameters that authors used to identify and classify gene duplication models.

Response: Done. 

Point 9: Please explain which Gene duplication event (tandem or segmental) contributed predominantly to the expansion of the WRKY genes and why. 

Response: Done. 

Point 10: Author mention in line 154-161 that they survey the SbWRKY expression profiles under different stress, but the results are not included. 

Response: We apologize if this section was not clear. We first downloaded public microarray datasets for sorghum, and then uploaded them to the Genevestigator platform for analysis. Figure 6 shows a heatmap representation of these results. 

Point 11: Authors should discuss more deeply all their results, for example, authors can discuss about the changes in the expression and if these changes can affect the plant development, or why one group of genes are more expressed in determinate stage, etc.

Response: Done. 

Point 12: Include more references throughout all the manuscript specifically in the discussion to increase the value of the paper. 

Response: Done. 

Point 13: Please identify the function of each motif identified by MEME and discuss about these functions, and how these motifs interact with the WRKY motif?

Response: Done for most motifs. We note however that some motifs have yet to be functionally characterized, which we mention in Results and in Discussion. 

Minor Comments:

Point 1: Authors should have one or more native English speakers review the manuscript before resubmission.

Response: Done. 

Point 2: In references and manuscript, please look carefully the gene and species names. They should be in italics. 

 Response: Done. 

Point 3: Please check grammar throughout the manuscript

Response: Done 

Response to Reviewer 3 Comments

We thank the reviewer for their thoughtful review and efforts towards improving our manuscript. 

Major points

Point 1: Figure 5C. Very good heat map representing WRKY expression under different environments. Please include what each environment represents in the figure legend to help readers understand gene expression under a specific stress environment.

Response: Done

Point 2: Table 1: Include the percentage of protein similarity between different WRKY genes with reference to previously characterized SbWRKY conferring seed shattering phenotype. This information will help readers understand structural variation between WRKY proteins. 

Response: Done. 

Point 3: Line 338: What is the rationale for selecting these five wrky genes for qPCR in your drought tolerance experiment? Include it in the M&M or discussion section.

Response: We now provide this information in the Discussion.

Point 4: Figure 2: In addition to phylogenetic analysis of sorghum and Arabidopsis WRKY genes, include synteny analysis of sorghum and Arabidopsis WRKY genes. This information will help on studying sorghum orthologs of previously characterized WRKY genes in model plants.

Response: This is an excellent suggestion, but we believe this falls outside of the scope of this current manuscript. However, we plan to start working on comparing synteny between sorghum, maize, rice and wheat with a focus on the WRKY gene family. 

Minor suggestions:

Lines 118 and 195: Change Morocoshi to Morokoshi

Response: Changed. 

Line 375: Missing reference

Response: Added. 

Abbreviate the gene names when used for the first time. SPL1 sweet potato and GAPDH

Table 1: WRKY88, include Sobic ID for this gene.

Response: Done.

---

## [Decision Letter · Decision Letter 1]

24 Jun 2020

PONE-D-20-02381R1

Genome-wide identification of WRKY transcription factor family members in sorghum (Sorghum bicolor (L.) moench)

PLOS ONE

Dear Dr. Elamin Hafiz Baillo,

Thank you for submitting your manuscript to PLOS ONE. After careful consideration, we feel that it has merit but does not fully meet PLOS ONE’s publication criteria as it currently stands. Therefore, we invite you to submit a revised version of the manuscript that addresses the points raised during the review process.

ACADEMIC EDITOR:

After careful review of the revised manuscript PONE-D-20-02381R entitled "Genome-wide identification of WRKY transcription factor family members in sorghum (Sorghum bicolor (L.) moench)", it is recommended for publication in PLOS ONE journal as research article. However, the authors are strongly suggested to bring the changed manuscript with the standard style of English presentation besides addressing to the reviewers' other comments.

.

We look forward to receiving your revised manuscript.

Kind regards,

Ramasamy Perumal, Ph.D.

Academic Editor

PLOS ONE

Reviewers' comments:

Reviewer's Responses to Questions

**Comments to the Author**

1. If the authors have adequately addressed your comments raised in a previous round of review and you feel that this manuscript is now acceptable for publication, you may indicate that here to bypass the “Comments to the Author” section, enter your conflict of interest statement in the “Confidential to Editor” section, and submit your "Accept" recommendation.

Reviewer #2: All comments have been addressed

2. Is the manuscript technically sound, and do the data support the conclusions?

Reviewer #2: Yes

3. Has the statistical analysis been performed appropriately and rigorously? 

Reviewer #2: I Don't Know

4. Have the authors made all data underlying the findings in their manuscript fully available?

Reviewer #2: Yes

5. Is the manuscript presented in an intelligible fashion and written in standard English?

Reviewer #2: Yes

6. Review Comments to the Author

Reviewer #2: 1. There are still some kind of problems with English usage and grammar throughout the manuscript, for example, Page 13 line 147, modify “were” for “was”. Authors should have one or more native English speakers review the manuscript before resubmission.

2. Please modify or edit the figure for Gene Ontology analysis, do not use the same figure that is given from the Blast2go software.

3. What authors refer to experiment 1 to 37 in figure 6C, please clarify and include the name for each experiment.

4. In particular I would like to commend the authors for the clarity of the writing principally in the results section.

5. Similarly, the discussion section would read better if some of the functional speculations were removed.

7. PLOS authors have the option to publish the peer review history of their article (what does this mean?). If published, this will include your full peer review and any attached files.

Reviewer #2: No

---

## [Author Response · Author response to Decision Letter 1]

6 Jul 2020

Dr. Ramasamy Perumal

Academic Editor

PLOS ONE

Dear Dr. Perumal,

We thank you and the reviewers for the helpful suggestions and comments. We have revised the manuscript and we used stander English as suggested, different native English speakers have reviewed the manuscript. We have responded to reviewer #2 comments and made the changes as requested. We hope you will find that this revised manuscript meets the requests of the Editor and reviewer #2, and is now acceptable for publication in PLOS ONE.

Please find our point-by-point responses to the reviewer #2 below.

Reminder:

We also would like to update our Funding Statement to reads as follows:

“This project was supported by The National Key Research and Development Program of China (2016YFD0100605), the Innovation Academy for Seed Design, Chinese Academy of Sciences; special exchange programme of the Chinese Academy of Sciences (category A).The authors sincerely thank the Center for Agricultural Resources Research, Institute of Genetics and Developmental Biology (CAS) for providing the facilities” 

Reviewer 2 Comments

We would like to thank the reviewer for their thorough review and efforts towards improving our manuscript. 

Point 1: There are still some kind of problems with English usage and grammar throughout the manuscript, for example, Page 13 line 147, modify “were” for “was”. Authors should have one or more native English speakers review the manuscript before resubmission.

Response: We have modified and made corrections throughout the manuscript, different native English speaker have reviewed the revised version as recommended.

Point 2: Please modify or edit the figure for Gene Ontology analysis, do not use the same figure that is given from the Blast2go software.

Response: We have edited the figure. 

Point 3: What authors refer to experiment 1 to 37 in figure 6C, please clarify and include the name for each experiment.

Response: We have clarified and included the name for each experiment in figure 6C. 

Point 4: In particular I would like to commend the authors for the clarity of the writing principally in the results section.

Response: We have made some changes and clarified.

Point 5: Similarly, the discussion section would read better if some of the functional speculations were removed.

Response: We have removed some functional speculations as suggested.

---

## [Editor Report · Decision Letter 2]

13 Jul 2020

Genome-wide identification of WRKY transcription factor family members in sorghum (Sorghum bicolor (L.) moench)

PONE-D-20-02381R2

Dear Dr. Elamin Hafiz Baillo,

We’re pleased to inform you that your manuscript has been judged scientifically suitable for publication and will be formally accepted for publication once it meets all outstanding technical requirements.

Kind regards,

Ramasamy Perumal, Ph.D.

Academic Editor

PLOS ONE

Additional Editor Comments (optional):

The authors made necessary changes in the revised version and hence recommended for publication in PLOS ONE as research article.
---

## [Editor Report · Acceptance letter]

15 Jul 2020

PONE-D-20-02381R2 

Genome-wide Identification of WRKY Transcription Factor Family Members in Sorghum (*Sorghum bicolor* (L.) moench) 

Dear Dr. Baillo:

I'm pleased to inform you that your manuscript has been deemed suitable for publication in PLOS ONE. Congratulations! Your manuscript is now with our production department. 

Kind regards, 

on behalf of

Dr. Ramasamy Perumal 

Academic Editor

PLOS ONE